# First Molecular Identification of *Fasciola gigantica* in Slaughtered Cattle in Cape Verde: Prevalence, Gross Pathological Lesions, Genetic Identification and Coprological Analysis

**DOI:** 10.3390/pathogens12010075

**Published:** 2023-01-03

**Authors:** Sara Levy, Manuela Calado, Teresa Letra Mateus, Madalena Vieira-Pinto

**Affiliations:** 1Veterinary Sciences Department, Universidade de Trás-os-Montes e Alto Douro, UTAD, Quinta de Prados, 5000-801 Vila Real, Portugal; 2Global Health and Tropical Medicine, GHTM, Instituto de Higiene e Medicina Tropical, IHMT, Universidade Nova de Lisboa, UNL, 1349-008 Lisbon, Portugal; 3CISAS Center for Research and Development in Agrifood Systems and Sustainability, Escola Superior Agrária, Instituto Politécnico de Viana do Castelo, Rua Escola Industrial e Comercial de Nun’Àlvares, 4900-347 Viana do Castelo, Portugal; 4EpiUnit Instituto de Saúde Pública da Universidade do Porto, Laboratory for Integrative and Translational Research in Population Health (ITR), 4050-091 Porto, Portugal; 5Veterinary and Animal Research Centre (CECAV), Universidade de Trás-os-Montes e Alto Douro, UTAD, Associate Laboratory for Animal and Veterinary Sciences (AL4AnimalS), 1300-477 Lisbon, Portugal

**Keywords:** fasciolosis, liver lesions, RFLP, active shedders, epidemiology

## Abstract

A study on fasciolosis prevalence, gross pathological lesions, fluke genetic identification and coprological analysis was carried out in slaughtered cattle from one abattoir in Cape Verde. Of the 131 cattle inspected over two months, 12 (9.0%) presented fasciolosis-compatible lesions (FCL) that resulted in liver condemnation. The genetic characterization of the flukes collected, through restriction fragment length polymorphism analysis of PCR-amplified fragments (PCR-RFLP), confirmed the presence of *Fasciola gigantica*; therefore, being the first identification of this species in cattle from Cape Verde. Animals that released *Fasciola* spp. eggs and, thus, responsible for environment contamination (positive shedders), were identified through coprological analysis (natural sedimentation technique). Of the 12 animals with FCL, samples from 11 were submitted to coprological analysis and 7 (63.6%) were found to be positive shedders. Furthermore, of the 82 animals with non-FCL, randomly selected for coprological analysis, 4 (4.9%) were also found to be positive shedders for *Fasciola* spp. The results of this study, regarding species identification and coprological analysis, are epidemiologically important to update the information regarding fasciolosis in Cape Verde. The new data could help implement effective strategies for disease control and mitigation, consequently reducing economic loss and the level of animal and human infection from the One Health perspective.

## 1. Introduction

Fasciolosis is a disease caused by members of the genus *Fasciola*, usually *Fasciola hepatica* or *Fasciola gigantica* [1].

It usually affects the bile ducts and liver parenchyma of ruminants, especially sheep, goats and cattle, the common definitive hosts. However, other important domestic and wild animals may also be affected: horses, donkeys, mules, and camelids [2]. Humans are also definitive hosts and play a significant part in the transmission of fasciolosis, especially in human hyperendemic zones [3]. The intermediate host is a snail (usually *Lymnaea natalensis* and *Lymnaea auricularia* for *F. gigantica* and *Lymnaea truncatula* for *F. hepatica*) [4]. The importance of the different intermediary hosts relies on their different geographical distribution due to climate (tropical vs. temperate) [5,6], temperature and type of habitat (deep and permanent water bodies vs. marshy areas that may dry occasionally) [5].

Recently Dermauw et al. (2021) stated that human fasciolosis is still a neglected disease and there is an urgent need to develop more epidemiological studies [7]. Human fasciolosis in Cape Verde has already been documented. The symptoms are usually fever, anorexia, severe abdominal pain, weight loss and hepatomegaly [8,9]. Cape Verdean citizens had to be evacuated to Portugal, France and to the United States due to acute symptoms, where they were diagnosed with fasciolosis [8,10,11]. Another case was of an American tourist that had spent some time in Cape Verde [8]. A Cape Verdean emigrant returned to the States, after being in his home country, and was immediately admitted to the hospital for evaluation of right upper quadrant tenderness, where he was later diagnosed [10].

On the other hand, studies on animal fasciolosis have been limited to an estimation of disease prevalence in live or slaughtered animals [12,13,14,15,16]. The causative species have only been identified morphologically—this is not reliable because parasites may differ slightly depending on the definitive host [17]. A distinction between the two species can be made accurately through molecular methods. This is important since the two species have different epidemiological characteristics (intermediary hosts, geographical areas, climates, and habitats) so disease understanding and control will differ.

Observation of the adult parasite inside the bile ducts of the liver through necropsy or post-mortem sanitary inspection at the slaughterhouse level is the simplest and most reliable method to diagnose fasciolosis [9]. So, slaughterhouse surveillance could be used to monitor the presence of the infections, estimate the prevalence, track the origin and study and implement a control program [18].

The detection of eggs in feces corroborates the animals’ role in the contamination and maintenance of the infection in the environment [19]. Furthermore, the spread of *Fasciola* spp. eggs through feces is a key point concerning the control of slaughterhouse effluents and wastewaters. Soil and water body contamination by slaughterhouse sludge presents an environmental and public health threat, not only regarding *Fasciola* spp. infections, but other parasites that do not need an intermediate host. Several pathogens can be found in this waste, such as parasites and their eggs, viruses, and bacteria [20]. Slaughterhouse employees are also at risk due to repeated contact and exposure to waste material [21].

The aims of this study were to determine the fasciolosis prevalence in cattle slaughtered in one abattoir in Santiago Island, Cape Verde; to characterize the level of hepatic lesions associated with fasciolosis; to evaluate the presence of *Fasciola* spp. eggs in slaughtered animals with and without fasciolosis compatible lesions (FCL) and characterize molecularly the *Fasciola* spp.; to reliably identify the *Fasciola* species present in Cape Verde and to exclude the possibility of hybrid existence.

## 2. Materials and Methods

The present study was undertaken in Santiago Island (991 km^2^, 266.161 inhabitants), Cape Verde, from January to February.

Cape Verde is an African country composed of ten islands on the Atlantic Ocean. The biggest and most populated island is Santiago, where the capital of the country, Praia, is located.

Cape Verde has a significant percentage of population living below the poverty line. Approximately 27% of the population lives in this situation, mainly in rural areas [22]. Historically, Cape Verde has always had enormous difficulties managing the lack of food resources due to the hard climate conditions. It imports around 80% of its food [23] but those goods are expensive to the average Cape Verdean citizen. Therefore, animal production is an important means of survival in Cape Verde, especially cattle, being the most common species seen in the abattoirs. Santiago is the island with the highest number of holdings per species [24].

Santiago has two abattoirs, Praia (capital) abattoir and Assomada abattoir. The first is located outside the city limits, in a region called S. Filipe. The second is in Assomada, a city right in the middle of Santiago.

This present study was undertaken in Praia abattoir. This abattoir comprises only one slaughter line officially used for three animal groups: bovine, swine and caprine. During this study, only bovines were slaughtered.

There was no treatment or control of effluents and waste in this facility. The content of the gastrointestinal tract was discharged directly into the sewage that was released into fields nearby. This included feces and digestive material.

For two months (January and February), meat inspection procedures were followed in the Praia abattoir. A total of 131 cattle were slaughtered. During this period animals with FCL were identified through a post-mortem inspection. It was considered an FCL positive case when fibrous and thickened biliary ducts and/or the presence of the fluke were observed [25,26]. Fecal samples were collected from animals with and without FCL (these were animals without any hepatic lesions and were randomly chosen) to explore the relationship between the presence of FCL and egg excretion. These samples were collected, individually, directly from their rectum to a sterile labelled plastic tube, containing formaldehyde at 10%, and stored at ambient temperature. Detected flukes were collected from the bile ducts to microtubes containing ethanol at 98% for later species identification.

For fecal analysis, 20 mL were taken from each fecal sample and later mixed with 50 mL of water. Water was further added to a total volume of 300 mL. The fecal solution was filtered, with the help of a metallic grid, and was left to deposit sediment for 20 min. The supernatant was discarded, and the process of sedimentation was repeated for 30 min, a total of four times. The final sediment was pipetted onto a microscope slide, to which a drop of methylene blue was added, and covered with a coverslip for microscope observation.

For total genomic DNA extraction, the DNA was isolated from flukes, according to Stothard et al. (1996) [27], with modifications as follows.

The posterior part of the fluke was homogenized in 600 μL lysis buffer and 10 μL of proteinase K and incubated at 55 °C for 60–90 min. Subsequently, 750 μL of chloroform isoamyl alcohol (24:1) was added and stirred for 2 min. The mixture was centrifuged (15 s at 13,800× *g*) resulting in an aqueous and an organic layer. The aqueous layer was collected and mixed with 1 mL of ice-cold absolute ethanol and centrifuged for 20 min at 13,800× *g*. The DNA pellet was washed in 500 μL of 70% ethanol and centrifuged for 10 min at 13,800× *g*, dried for 10 min at 55 °C and re-dissolved in 45 μL of TE buffer and stored at −20 °C.

The PCR was used to amplify and generate copies of a specific and well-known DNA sequence of the parasite, to later identify the species. The region amplified was the ITS1 region.

This region was amplified using primers *FascR* and *FascF* [28]. For each sample, 1 μL of each primer, 1 μL of DNA and 22 μL of double distilled water, were added to a PCR bead (IllustraTM PuReTaq Ready-To-Go PCR Beads, GE Healthcare). Amplifications were carried out in a thermocycler (Aviso, GmbH Mechatronic Systems) with the following conditions: 5 min at 95 °C, followed by 30 cycles at 95 °C for 45 s, 60 °C for 45 s, 72 °C for 1 min and a final step at 72 °C for 7 min.

Samples were prepared with 2 μL of PCR product and 1.5 μL of loading buffer (Crystal 5× DNA Loading Buffer Blue, Bioline, Camarillo, CA, USA).

The samples were analyzed through electrophoresis at 120 V, in a 1.5% agarose gel in TAE buffer (40 mM Tris-acetate, pH 8.0, 1 mM EDTA), with EtBr stain.

The visualization of the DNA migration was under ultraviolet light on a transilluminator (Alphamanager HP, Alpha Innotech, San Leandro, CA, USA).

For restriction fragment length polymorphism (RFLP), PCR products were cut with *Tas*I (FastDigest Tsp509I, Fermentas, Waltham, MA, USA) [28] by adding 8 µL of double distilled water, 0.5 µL of *Tas*I, 1.5 µL of buffer (10× FastDigest Green Buffer, Fermentas) and 5 µL of PCR product, in a total volume of 14 µL, and incubated at 65 °C for 2 h and 50 min followed by 20 min at 95 °C. Restriction fragments were separated as described above, in 2% agarose.

## 3. Results

### 3.1. Prevalence, Gross Pathological Lesions, and Presence of Flukes

During the study period, 131 slaughtered cattle were inspected at the abattoir. From those, 12 (9.0%) presented liver lesions compatible with *Fasciola* spp. infection—FCL.

This was the only cause of offal/meat rejection observed during the study period. No other lesions or diseases were detected on the carcass or offal.

Most of the lesions occurred in the left lobe of the liver (8 out of 12) (Figure 1). In most cases, there was hepatomegaly, with a round appearance and at times, the liver was heterogeneous in color, with pale areas alternating with dark ones. The most common lesion observed was fibrous and thickened biliary ducts (Figure 2) that were, occasionally, clearly visible on the surface of the organ, like a tunnel of yellow scar tissue (Figure 1). The presence of flukes in the bile ducts was observed in 11 of the 12 cases (91.7%) (Figure 3). From those, eight had more than one fluke present. In total, 19 parasites were collected for further analysis.

### 3.2. Coprological Analysis

*Fasciola* spp. eggs were detected in 7 of the 11 collected fecal samples (63.6%) from FCL-positive animals (Table 1).

### 3.3. Genetic Analysis and Species Identification

All 19 flukes collected were genetically analyzed. The ITS region was amplified by PCR for each fluke, resulting in the expected fragment of 500 bp for all samples except for samples 4 and 13. RFLP with *Tas*I, produced fragments similar to the expected sizes of 93,151 and 219 bp [28], reliably confirming *F. gigantica* (Figure 4).

## 4. Discussion and Conclusions

We estimated the prevalence of fasciolosis in slaughtered cattle during post-mortem inspection in the Praia slaughterhouse, the capital of Santiago Island, Cape Verde. The prevalence found (9.0%) was considerably lower than the one previously estimated by Rosa et al. (2004) [12] (39.2%) and Centeio, 2008 [15] (20.0%). The first study was conducted in the Praia abattoir in the period (1994–1999) when there was no abattoir operating in Assomada, so production was at its peak, and animals from all Santiago’s counties were slaughtered in the Praia abattoir. The second study was carried out in the Assomada abattoir, the island’s main abattoir. These differences in prevalence could, partially, be explained by differences in the slaughtered animals’ origins (that currently cannot be traced), as fasciolosis prevalence may vary throughout Santiago Island [29].

The season of the year may also be responsible for discrepancies. In Cape Verde, there are two seasons: the rainy season, from August to October, and the dry season, from December to June [30]. The studies conducted by Rosa et al. (2004) [12] were focused on two periods: October and November (from the rainy to the dry season), where the prevalence found was 52.38%, and March and April (full dry season), where the prevalence was lower (36.36%). Our work was carried out in January and February, the dry season, possibly, during a period of lower prevalence.

According to EFSA (2013) [31], several studies indicated that fasciolosis in cattle is underdiagnosed by clinical surveillance, its prevalence being more accurately evaluated during post-mortem inspection. In fact, slaughterhouse records present an important source of information that could/should be used to provide an effective monitoring system (passive), allowing periodic reports to human health and veterinarian authorities. Nevertheless, no monitoring or reporting system exists in Cape Verde. These systems, through the identification of the geographical distribution of zoonotic diseases and the identification of hot spots could support the development and implementation of control programs that c attempt to reduce risk factors and recommend the use of drugs in a more strategic way [32].

From the 11 analyzed fecal samples from FCL positive animals, 7 (63.6%) revealed the presence of *Fasciola* spp. eggs. The failure to detect eggs could have occurred due to errors regarding the sedimentation technique, failure in the egg’s visualization or absence of eggs in the fecal sample analyzed or being in an early stage of disease. Weak or early stages of infection can pass undetected since the flukes are required to be sexually mature for the eggs to be visualized in the feces of the hosts [33]. Non-FCL cases (4 in 82) also presented *Fasciola* spp. eggs in feces. This data showed that animals without any lesions and considered negative at meat inspection were actually infected, resulting in false-negative cases. The prevalence of bovine fasciolosis based on meat inspection can be underestimated due to the low sensitivity of this procedure [34] but liver inspection sensitivity could be increased if livers were effectively sliced [35].

These results should be taken into consideration during the implementation of any passive monitoring systems at slaughterhouse level based on liver evaluation.

Several authors highlighted the importance of human fasciolosis in Cape Verde, making this zoonotic disease a public health issue [13,14]. As stated by Mas-Coma et al. (1999) [3], the importance of human fasciolosis has been proved by the increasing number of cases over the years. This zoonotic disease is not transmissible by the ingestion of cattle meat or offal. It occurs by ingestion of vegetables with metacercaria of *Fasciola* spp. [36]. One example in Cape Verde, is of a man diagnosed with fasciolosis, after consuming home-grown watercress. [10]. The evidence of parasite’s eggs in the feces of slaughtered animals in Praia slaughterhouse, and the lack of treatment and control of effluents and waste in this facility, are indicators of a threat to public health. Animal feces from the slaughterhouse were discharged directly into the environment, without treatment, on fields that surrounded the building. This action was carried out by the slaughterhouse employees, possibly allowing the perpetuation of the parasite’s cycle (due to the probable presence of the intermediate host in the fields surrounding the slaughterhouse), putting animals that graze near the slaughterhouse at danger, as well as threatening public health, since there was agriculture production nearby. Historically, Cape Verde has always had enormous difficulties managing the lack of food resources due to the hard climate conditions; it imports around 80% of its food [23]. Approximately 27% of the Cape Verdean population lives below the poverty line, mainly in rural areas, where a lack of basic sanitation, electricity and water resources is a well-set reality [22], turning to homegrown produce and cattle production as means of sustenance. During a trip to a village near Praia (S. Domingos) at the same time of the study period, it was observed how the transmission dynamic could be at play. Cattle was seen grazing and defecating near water courses (Figure 5) where *Lymnaea* spp. was suspected to be present (Figure 6). The water was pumped to irrigate horticultural production nearby (Figure 7).

The parasites collected were identified as *F. gigantica* by PCR-RFLP of the ITS region. This is the first report of *F. gigantica* in Cape Verde being reliably identified by molecular means. According to the authors’ knowledge, the confirmation of the species by genetic means has never been carried out before in this country. There are several published studies [12,14,16] mentioning the presence of *F. gigantica* and its intermediary host (*L. natalensis*), in the country, but this identification was carried out through means of observation and suspicion. Morphological identification can be reportedly difficult and does not guarantee authenticity. It is crucial to distinguish between *F. hepatica* and *F. gigantica* due to differences in the intermediate host, control strategies, pattern of pathologic effects and epidemiological characteristics—hence the importance of this new obtained data.

Another important aspect regarding the molecular study was the possibility of identifying hybrids. Since Cape Verde imports some of its cattle from countries where *F. hepatica* prevails, such as Portugal, it was relevant to see if a hybrid had developed. Hybrid existence has already been reported in countries such as Japan [37,38], Korea [39], China [40], Vietnam [41], Egypt [42], Bangladesh [38] and Chad [43]. Furthermore, in a study from Hasanpour [44], regarding the molecular characterization of *Fasciola* spp. from some parts of Iran, it was found that there is a substantial genetic difference between the *F. gigantica* populations of Asia and Africa [44].

The presence of fasciolosis was the only cause for meat/offal condemnation seen in this study. This condition elevates the problem even more, since this disease leads to economic loss regarding meat production, such as liver condemnation [15] and carcass weight loss and emaciation [45]. Even in live animals, fasciolosis decreases milk production and animal fertility [46].

As a conclusion, we reiterate the importance of implementing a passive fasciolosis monitoring system at slaughterhouse level, which should be coupled with an animal identification system in the country, to trace back the infection to its origin and act accordingly so as to enable control measures. Thus, animal and human *Fasciola* spp. infections can be decreased and prevented under the One Health approach.

## Figures and Tables

**Figure 1 pathogens-12-00075-f001:**
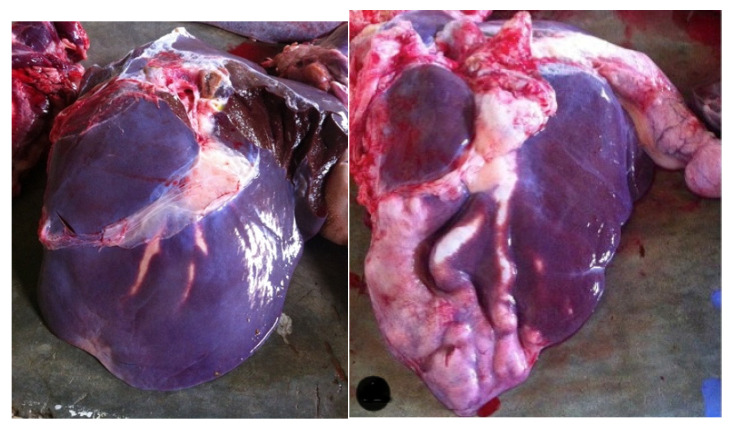
Two bovine livers with bile duct thickening on the left lobes of slaughtered cattle from Cape Verde.

**Figure 2 pathogens-12-00075-f002:**
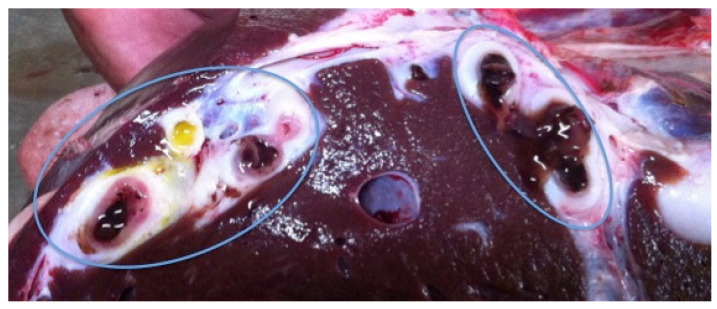
Hyperplastic bile ducts of a slaughtered cattle from Cape Verde.

**Figure 3 pathogens-12-00075-f003:**
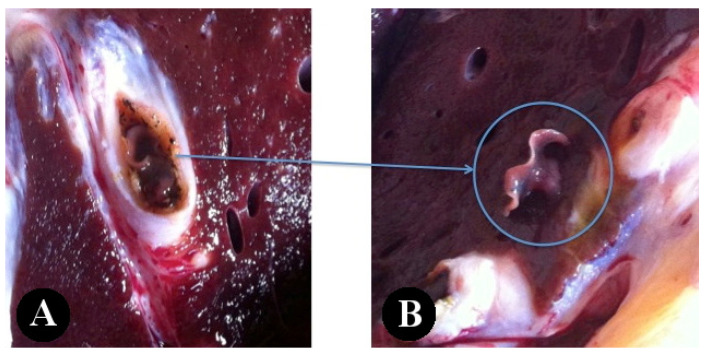
*Fasciola* spp. inside the hypertrophied bile duct (**A**). *Fasciola* spp. moving on the surface cut (**B**) of a slaughtered bovine from Cape Verde.

**Figure 4 pathogens-12-00075-f004:**
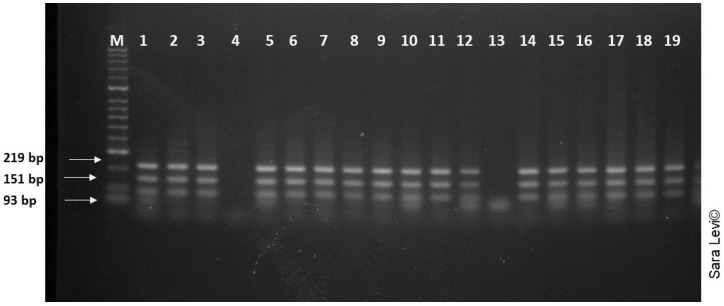
Agarose gel electrophoresis of the amplified ITS 1 region from the 19 samples, after digestion with *Tas*I (FastDigest Tsp509I, Fermentas). M: 2000 bp molecular weight marker. N: negative control.

**Figure 5 pathogens-12-00075-f005:**
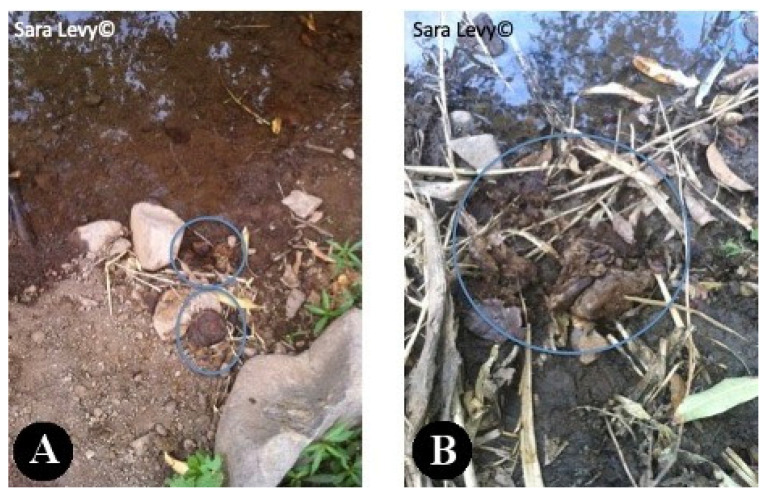
Bovine feces near a watercourse (**A**,**B**) in S. Domingos, Santiago.

**Figure 6 pathogens-12-00075-f006:**
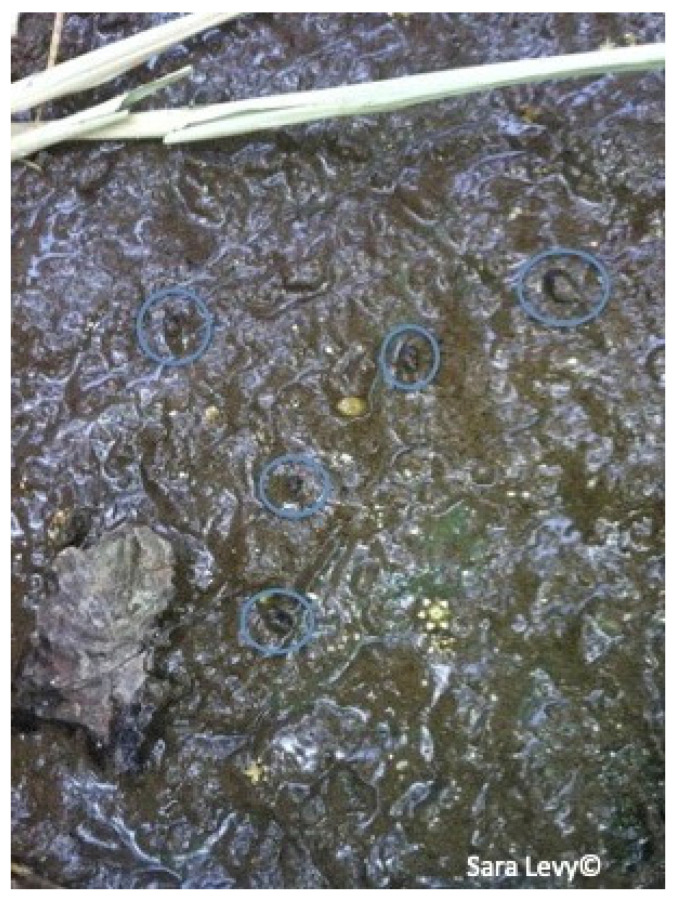
*Lymnaea* spp. near a watercourse in S. Domingos, Santiago.

**Figure 7 pathogens-12-00075-f007:**
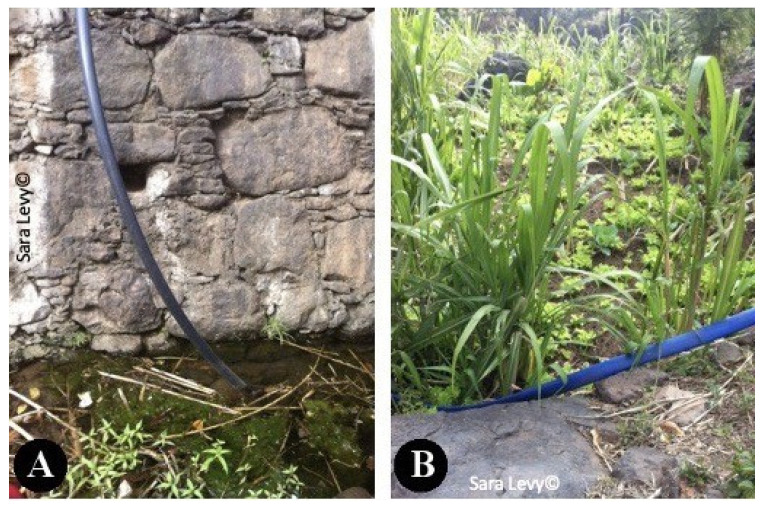
Water being pumped from a watercourse (**A**) into a horticultural production (**B**), in S. Domingos, Santiago.

**Table 1 pathogens-12-00075-t001:** Record of *Fasciola* spp. eggs in feces of animals with fasciolosis-compatible lesions (FCL) and presence of flukes inside the bile ducts of slaughtered cattle from Cape Verde.

FCL Case Number	Liver Analysis (Presence of Flukes)	Fecal Analysis (Presence of *Fasciola* spp. Eggs)
1	1	-
2	1	1
3	1	1
4	1	0
5	1	1
6	1	1
7	0	0
8	1	0
9	1	1
10	1	0
11	1	1
12	1	1
**Total**	**11**	**7**

0: Absence; 1: presence; -: not analyzed. From the 82 fecal samples of the non-FCL cases randomly collected and analyzed, 4 also presented *Fasciola* spp. eggs.

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
