# Peer review of "First Molecular Identification of Fasciola gigantica in Slaughtered Cattle in Cape Verde: Prevalence, Gross Pathological Lesions, Genetic Identification and Coprological Analysis"

_pathogens, 2023, doi:10.3390/pathogens12010075_

Round 1

Reviewer 1 Report

The article exposes an investigation of relevance to the area, and uses appropriate methodology.

However, the results presented in table 1 are only limited to showing the presence/absence of fasciola or eggs.

It would be important to modify that information and show quantitative data that may be of importance in the future if it is possible to face some type of tracking of the origin of the animals.

Author Response

Dear reviewer

Thank you for considering our manuscript for publication in Pathogens. We acknowledge the valuable feedback provided it, contributing to improve the manuscript. We describe our responses below.

Madalena Vieira-Pinto

  Authors' responses to reviewer 1 comments

  • The article exposes an investigation of relevance to the area, and uses appropriate methodology.

Answer: The authors are grateful for the kind comments sent by the reviewer.

  • However, the results presented in table 1 are only limited to showing the presence/absence of fasciola or eggs. It would be important to modify that information and show quantitative data that may be of importance in the future if it is possible to face some type of tracking of the origin of the animals.

Answer: The authors thank the suggestions. Concerning the number of parasites present in the liver, accounting has not been done as it would not be possible at the slaughterhouse context. Besides, our aim was not to quantify the burden of infection but solely the assess if the risk was present or not. As it is a zoonotic parasite, the presence of only one parasite is enough to confirm the zoonotic risk. About the eggs, the methodology used was qualitative for the same reasons stated before, our aim was not to asses the burden of infection in animals but the presence of potential zoonotic risk.The table shows the presence/absence of Fasciola eggs in faeces and adult parasites in the liver of the 11 FCL positive cases – the cases that had hepatic lesions compatible with Fasciola spp. infections. Unfortunately, during the study, the number of eggs in the faecal matter wasn’t quantified neither the total number of flukes in the organ of each animal. The main goal of the table is to correlate these two data and show which cases are positive shedders. Each case has a number attributed  (FCL case from 1 to 11) and the date that the samples (faecal and adult parasites) were collected (that correlates to the slaughter date) that could allow some sort of tracking.

Reviewer 2 Report

- Why 

- Line 12-13: Formatting needs revision.

- Line 85: Materials and Methods should be the item "2".

- Figure 1, 2 and 3 Legends should be improved with "of a slaughtered cattle from Cape Verde".

- Table 1 Legend should be improved with "of slaughtered cattle from Cape Verde".

- Figure is not in the main text.

- Line 203-211: The sentence "The second study was developed in Assomada abattoir, the island’s main abattoir. These prevalence differences could, partially, be explained by differences in the slaughtered animals’ origin (that currently cannot be traced), as fasciolosis prevalence may vary throughout Santiago Island [29]. Unfortunately, at present, there is no animal identification system in Cape Verde to test this hypothesis. This subject constituted the study’s biggest limitation. The individual that takes the animals to be slaughtered is not their original owner. The animals are usually bought the day before, at a local fair, and are not registered nor have any documents providing their origin or breeder’s information. For that reason, they arrive at the slaughterhouse without any documents, tattoos, or earrings – no information regarding their country of origin." is not a scientific discussion, it is just a speculation and is not a possible conclusion of the study. It must be withdrawn.

- Line 261, 262, 267 and 271: Fasciola genus name must be abbreviated.

- About the discussion, some important aspects need to be considered: First, the discussion is confusing and repetitive in certain parts, so that the text can be reorganized in a more concise way and with fewer paragraph breaks; According to many studies describe the occurrence of Fasciola gigantica in cattle in Cape Verde, the present study being the first description based on molecular biology. This point needs to be made clear both in the title and given more emphasis in the discussion; Finally, the approach to One-Health in the discussion needs to make it much clearer that infection in humans occurs through ingestion of metacercariae in vegetation, exploring this dynamic and relevance in Cape Verde.

- In the references, there are many citations to monographs and course conclusion works. Replace these references with published and easily found scientific articles, so that they can be used by the scientific community.

Author Response

Dear reviewer

Thank you for considering our manuscript for publication in Pathogens. We acknowledge the valuable feedback provided it, contributing to improve the manuscript. We describe our responses below.

Madalena Vieira-Pinto

  Authors' responses to reviewer 2 comments

- Line 12-13: Formatting needs revision.

Answer: Thank you. This was changed accordingly.

- Line 85: Materials and Methods should be the item "2".

Answer: Changed accordingly.

- Figure 1, 2 and 3 Legends should be improved with "of a slaughtered cattle from Cape Verde".

Answer: Changed accordingly.

- Table 1 Legend should be improved with "of slaughtered cattle from Cape Verde".

Answer: Changed accordingly.

- Figure is not in the main text.

Answer: Added accordingly.

- Line 203-211: The sentence "The second study was developed in Assomada abattoir, the island’s main abattoir. These prevalence differences could, partially, be explained by differences in the slaughtered animals’ origin (that currently cannot be traced), as fasciolosis prevalence may vary throughout Santiago Island [29]. Unfortunately, at present, there is no animal identification system in Cape Verde to test this hypothesis. This subject constituted the study’s biggest limitation. The individual that takes the animals to be slaughtered is not their original owner. The animals are usually bought the day before, at a local fair, and are not registered nor have any documents providing their origin or breeder’s information. For that reason, they arrive at the slaughterhouse without any documents, tattoos, or earrings – no information regarding their country of origin." is not a scientific discussion, it is just a speculation and is not a possible conclusion of the study. It must be withdrawn.

Answer: Sentence has been removed accordingly.

- Line 261, 262, 267 and 271: Fasciola genus name must be abbreviated.

Answer: Changed accordingly.

- About the discussion, some important aspects need to be considered:

- First, the discussion is confusing and repetitive in certain parts, so that the text can be reorganized in a more concise way and with fewer paragraph breaks;

Answer: Changed, hopefully it makes more sense now.

- According to many studies describe the occurrence of Fasciola gigantica in cattle in Cape Verde, the present study being the first description based on molecular biology. This point needs to be made clear both in the title and given more emphasis in the discussion;

Answer: Some wording changed and extra info added.

- Finally, the approach to One-Health in the discussion needs to make it much clearer that infection in humans occurs through ingestion of metacercariae in vegetation, exploring this dynamic and relevance in Cape Verde.

Answer: We thank the reviewer for this comment. This issue is clearly addressed in lines 249-250 and then explored concerning Cape Verde reality in lines 251 – 260. Nevertheless, more info added accordingly.

- In the references, there are many citations to monographs and course conclusion works. Replace these references with published and easily found scientific articles, so that they can be used by the scientific community.

Answer: As mentioned in the article, Cape Verde is considered a developing country with significant poverty. All the monographs and master thesis on the references of this work are authored by Cape Verdean students finishing their bachelor’s or master’s degrees - these were nevertheless carried out and then examined and evaluated by a panel of Professors /experts. It is extremely challenging for these students to formally publish any of their work due to economic and linguistic barriers. Because of the specific nature of our work /study and bearing in mind the country size and its population, works/published data in this field are scarce or old-dated, therefore we must rely on this grey literature to give some introduction to our work. By publishing new data, even relying on monographs/master theses for background info, we can, hopefully, significantly update the amount of data on the disease in order to improve the One-Health and encourage new studies to be carried out and published.

Round 2

Reviewer 2 Report

- Line 84: Despite being described in the abstract, the FCL acronym was not described in the main text. As this is the first appearance of the acronym in the main text, it should be spelled out here.

- Line 88: the term "km2" must have the number "2" superscripted.

- In Table 1 the FCL acronym was not described.

- In Table 1 replace "spp" with "spp.".

- Line 264: replace "spp" with "spp."

- Line 270: Lymnaea must be in italic.

- Line 278: Fasciola genus must be abreviatted.

- Line 281: Lymnaea genus must br abreviatted.

- Line 284: Fasciola genus must be abreviatted.

- Line 292: Hasanpour et al. (2020) it is out of standard citation.

Author Response

Dear Reviewer

Thank you for considering our manuscript for publication in Pathogens. We acknowledge the valuable feedback provided by the Reviewer, contributing to improve the manuscript. We describe our responses below.

Madalena Vieira-Pinto

Authors' responses to reviewer 2 comments

- Line 84: Despite being described in the abstract, the FCL acronym was not described in the main text. As this is the first appearance of the acronym in the main text, it should be spelled out here.

Answer: Changed accordingly.

- Line 88: the term "km2" must have the number "2" superscripted.

Answer: Changed accordingly.

- In Table 1 the FCL acronym was not described.

Answer: Changed accordingly.

- In Table 1 replace "spp" with "spp."

Answer: Changed accordingly.

- Line 264: replace "spp" with "spp."

Answer: Changed accordingly.

- Line 270: Lymnaea must be in italic.

Answer: Changed accordingly.

- Line 278: Fasciola genus must be abreviatted.

Answer: Changed accordingly.

- Line 281: Lymnaea genus must br abreviatted.

Answer: Changed accordingly.

- Line 284: Fasciola genus must be abreviatted.

Answer: Changed accordingly.

- Line 292: Hasanpour et al. (2020) it is out of standard citation.

Answer: Changed accordingly.
